Subject Category:
Biology (whole organism)

Subject Areas:
behaviour

Keywords:
hybrid, genomic imprinting,
maternal priming, *Mus musculus domesticus*,
*Mus spretus*, pup retrieval

Author for correspondence:
Sarah Gardner
e-mail: sagardner11@gmail.com

# Placental genotype affects early postpartum maternal behaviour

Sarah Gardner[1,2], Jennifer L. Grindstaff[1]
and Polly Campbell[1,2]

[1]Department of Integrative Biology, Oklahoma State University, Stillwater, OK, USA
[2]Department of Evolution, Ecology, and Organismal Biology, University of California Riverside, Riverside, CA, USA

 SG, 0000-0002-2861-5424

The mammalian placenta is a source of endocrine signals that prime the onset of maternal care at parturition. While consequences of placental dysfunction for offspring growth are well defined, how altered placental signalling might affect maternal behaviour is unstudied in a natural system. In the cross between sympatric mouse species, *Mus musculus domesticus* and *Mus spretus*, hybrid placentas are undersized and show misexpression of genes critical to placental endocrine function. Using this cross, we quantified the effects of placental dysregulation on maternal and anxiety-like behaviours in mice that differed only in pregnancy type. Relative to mothers of conspecific litters, females exposed to hybrid placentas did not differ in anxiety-like behaviours but were slower to retrieve 1-day-old pups and spent less time in the nest on the night following parturition. Early deficits in maternal responsiveness were not explained by reduced ultrasonic vocalization production in hybrid pups and there was no effect of pup genotype on measures of maternal behaviour and physiology collected after the first 24 h postpartum. These results suggest that placental dysregulation leads to poor maternal priming, the effect of which is alleviated by continued exposure to pups. This study provides new insight into the placental mediation of mother–offspring interactions.

## 1. Introduction

Mammalian maternal care is unique in that mothers have pre- and postnatal obligate investments: gestation and lactation. The developmental environment that a mother provides—during gestation and postpartum—has direct consequences for the growth and cognitive development of her offspring. Variation in pre- and postpartum maternal behaviour can lead to long-lasting epigenetic modifications in offspring genomes [1–3]. These modifications can

affect offspring physiology, behaviour and fitness [4–6]. For example, rat pups that receive lower levels of maternal care display increased anxiety-like behaviours and impaired spatial learning and memory as adults [7]. Just as mothers directly shape offspring development, offspring can affect maternal physiology and behaviour during gestation. Signals from offspring promote increased maternal food intake throughout pregnancy [8] and prime females to provide necessary care at birth [9].

These reciprocal interactions between mothers and offspring are mediated through the placenta, a temporary endocrine organ that forms in the uterus from the trophoblast cells of the developing embryo [10]. Aside from maternal blood vessels and the thin layer of cells that comprise the maternal decidua, the mature placenta is embryo-derived. Best known for its function in nutrient transfer from mother to embryo, the placenta is also a major source of hormones. In rodents, placental lactogens are the most numerous; 22 placental lactogens are highly expressed in the mouse placenta during mid–late gestation and are secreted into the maternal bloodstream [11–13]. Although the cellular targets of most placental lactogens are unknown, a subset binds to prolactin receptors in the maternal hypothalamus, acts to promote food intake and, in tandem with maternal hormones, primes the onset of maternal care at parturition [11,14–16].

Placental development and endocrine function are, in part, controlled by imprinted genes. Classified by their unique mode of expression, imprinted genes are autosomal genes with monoallelic expression. Depending on the gene, either the maternal or paternal allele is expressed, while the other is suppressed by DNA methylation, long non-coding RNAs or other epigenetic regulatory mechanisms [17]. Of the approximately 150 validated imprinted genes in mouse and human [18], the majority are highly expressed in the placenta and the brain. Correct dosage of imprinted genes in the placenta is critical to normal growth and development of the conceptus (the collective term for the fetus and the placenta) [19]. Because imprinted genes are highly expressed in the placental endocrine compartment, the source of placental lactogens that target the maternal brain [20], their potential to modulate maternal physiology and behaviour is considerable [10,15].

To date, the reciprocal effects of imprinted genes on offspring and mothers have been demonstrated using single-gene manipulations in laboratory mice. The best-studied example, *Peg3* (paternally expressed gene 3), is co-expressed from the paternally inherited allele in the maternal hypothalamus and the placenta and affects maternal behaviours [21]. Females with silenced *Peg3* exhibit increased latencies in pup retrieval, decreased licking and grooming of pups, decreased time spent nursing pups and impaired milk let-down [22]. These behavioural and physiological changes lead to impaired growth and development in offspring [23,24]. Notably, maternal behaviours also decrease when *Peg3* is inactive only in the placenta and fetal hypothalamus, suggesting that normal expression of *Peg3* during conceptus development is necessary for the induction of maternal behaviours [25–27]. Thus, both instances—either mothers or offspring without *Peg3*—lead to deficits in maternal behaviours. Other examples of imprinted genes that influence maternal physiology and behaviour include maternally expressed *Grb10* and *Phlda2*. *Grb10* impacts both conceptus growth and postnatal resource provisioning [28] whereas *Phlda2* negatively regulates the proliferation of placental endocrine cell lineages, with secondary effects on placental hormone production [29]. Notably, loss of function of *Phlda2* in the fetally derived placenta results in increased maternal nurturing by wild-type dams whereas loss-of-imprinting of *Phlda2* (increased expression) results in increased focus on nest building, a non-pup-directed behaviour [30].

Collectively, these studies demonstrate that imprinted genes and maternal behaviours are interconnected beyond a mother's own genes [31,32]. However, while single-gene manipulations can reveal the function of individual imprinted genes, using a natural hybrid system (i.e. species that occasionally hybridize in nature [33]) in which multiple genes are transgressively misexpressed, provides a more holistic view of the impact of imprinted genes on maternal behaviours. In the hybrid system used in the experiment described here, placental misexpression of imprinted genes and placental lactogens is correlated with altered maternal gene expression in the medial preoptic area of the hypothalamus [34]. Placental growth effects associated with abnormal expression of imprinted genes have been well characterized in a number of rodent genera, including *Mus* [35–38]. Yet, the opportunity these systems provide to test for effects of placental dysregulation on the behaviour of otherwise normal mothers has gone unrecognized.

In the cross between house mouse subspecies, *Mus musculus domesticus*, and sympatric congener, *M. spretus*, hybrid conceptuses are oversized when the mother is *M. spretus* and undersized when the mother is *M. m. domesticus* [38,39]. Here, we focus on the latter direction of the cross, in which multiple lines of evidence suggest that placental dysregulation should negatively impact maternal behaviours. First, size reduction in hybrid placentas is particularly pronounced in the endocrine compartment [40].

Second, the expression of placental lactogens, together with other placental gene families important to maternal–fetal interactions, is significantly reduced in hybrid relative to both *M. m. domesticus* and *M. spretus* placentas [34]. Third, placental overexpression of *Phlda2* reduces several pup-directed maternal behaviours in wild-type laboratory mice [30], and *Phlda2* is among the imprinted genes with significant overexpression in hybrid placentae [34]. Additionally, overexpression of *Phlda2* decreases the size of the endocrine compartment of the placenta [29]. Fourth, female *M. m. domesticus* carrying near-term hybrid litters have altered expression in the hypothalamus relative to conspecific mothers of normal litters [34]. In particular, *Drd3*, a dopamine receptor associated with treatment-resistant major depression in humans [41] and anxiety- and depressive-like behaviours in a laboratory mouse knockout [42], is downregulated in the maternal brain in hybrid pregnancies [34].

Here, we test for behavioural effects of exposure to abnormal placental signals in *M. m. domesticus* mothers of hybrid relative to conspecific litters. Motivated by altered *Drd3* expression in maternal brains, we tested for evidence of higher anxiety in mothers of hybrids during late gestation. Maternal behaviours were measured during the first 5 days postpartum, when high levels of maternal care are most critical to pup survival [43]. Given the evidence that hybrid placentae produce weaker signals that prime the onset of maternal care [30], we expected that mothers of hybrids would take longer to retrieve pups and would spend less time in the nest than mothers of conspecific litters. Coordination between maternal physiology and pup behaviour was evaluated with a suckling assay. We expected that mothers of hybrids would gain less weight when separated from their pups (an indication of less milk production), and that hybrid pups would regain less weight following a reunion with their mothers. Collectively, the results of this study provide new insight into the effects of unborn offspring on mothers, and the consequences of disrupted genomic imprinting and placental dysregulation for mother–offspring interactions.

# 2. Material and methods

## 2.1. Mouse husbandry and cohort information

The wild-derived inbred mice used in this study were maintained on a 12 L : 12 D cycle with lights on at 09.30 and were provided with ad libitum food (LabDiet® 5001 Rodent Diet) and water. *M. m. domesticus* was represented by the WSB/EiJ strain (Jackson Laboratory) and *M. spretus* was represented by the SFM/ Pas strain (Montpellier Wild Mice Genetic Repository). All animal procedures were approved by the Oklahoma State University IACUC under protocol # AS-1-41.

Adult (mean age ± s.d.: 116 ± 54 days, $n = 33$) female *M. m. domesticus* were paired to either a male *M. m. domesticus* or male *M. spretus* (163 ± 82 days, $n = 33$) for 14 days. Each pair was given a cotton nestlet and paper hut and was left undisturbed until male removal. Females were monitored daily for pregnancy and parturition an additional 14 days after male removal. Females used in this study were either first (28/41 litters), second (11/41 litters) or third (2/41 litters) time mothers. Seven females produced two litters that were both used, and one female produced three litters that were all used. A total of 19 hybrid litters were produced and 22 conspecific litters were produced. Average litter size did not differ between groups (conspecific: 3.78 ± 0.27 pups, hybrid: 3.45 ± 0.23 pups; $t = -0.922$, d.f. = 40.74, $p = 0.362$). Because a subset of all mothers was used in each behavioural assay, we also used Levene's test to check for unequal variance in litter size between mothers of conspecific and hybrid litters in a given assay. In all cases, the test was non-significant, indicating equal variance across the two groups.

## 2.2. Open-field trials

The open-field arena comprised a 16-square grid enclosed by a clear Plexiglas box (60.96 × 60.96 × 60.96 cm) with no top. Trials were run in the light cycle (85 lux) between 10.00 and 13.00 h and were recorded with a Panasonic® HC-W850 camcorder positioned above the arena. We aimed to match female gestational day to that in our prior study of expression in the maternal hypothalamus (day 18 of 21) [34]. Females used in the assay were estimated to be in late gestation based on visual assessment and date of pairing; actual gestational day (mean 18 ± s.d. 1.4, range: 15–21) was back-calculated after parturition. At the start of each trial, the mouse was placed in the middle of the apparatus in an opaque PVC cylinder. The cylinder was removed within 10–20 s and the trial was run for 5 min, starting when the mouse first entered the outer edge of the grid. The apparatus was cleaned with 70% ethanol before each trial. Trials were scored for number of lines crossed (a measure of activity and exploration), and latency to first enter the central

squares in the grid and total time in centre (measures of anxiety-like behaviours, where longer latencies and less time in the centre are proxies for a more anxious phenotype). Because some individuals showed pronounced freezing upon the removal of the PVC cylinder, latency to leave the centre of the apparatus at the beginning of the trial was also scored.

## 2.3. Pup retrieval

Pup retrieval tests were conducted within 24 h of parturition, during the light cycle between 13.00 and 17.00 h. To begin the test, the mother was removed from the home cage for 30 s. Three pups were removed from the nest and placed at different, equidistant points away from the nest. The mother was then reintroduced to the cage and scored for latency to return each of the pups to the nest. If the mother did not retrieve all pups within 15 min (latency = 900 s), the test was terminated [22]. Females included in the analysis all retrieved at least one pup in the test. All trials were videotaped using a Panasonic® HC-W850 camcorder and scored post-trial. Litters were only used in this behavioural test if the number of pups in the litter was three or more; additional pups not used in the test were placed under a heat lamp and returned to the home cage at the end of the test.

## 2.4. Home-cage activity

Home-cage activity levels of females with pups were monitored continuously from parturition for 96 h (termination was at the onset of the light cycle 5 days after parturition) using an automated monitoring system that recorded the number of times a female crossed an infrared beam per unit time (VitalView Animal Monitoring Software, Version 5.0). We used activity as a proxy for the relative amount of maternal care each female provided, with higher activity indicative of a female spending less time in the nest with her pups.

## 2.5. Suckling/milk let-down

On postnatal day 5, suckling by pups and maternal milk let-down were measured. Our assay design followed that of Curley and colleagues [25]. One hour after the beginning of the dark cycle, mother and pups were weighed individually and all pups within a litter were placed in a holding cage under a heat lamp in a separate room from their mother. After 2 h, mother and pups were reweighed and pups were returned to their home cage. The pups and mother were then weighed every hour for 4 h following reunion [25].

## 2.6. Ultrasonic vocalizations

Neonatal rodents displaced from their nest produce ultrasonic vocalizations (USVs) that promote maternal localization and retrieval [44]. Because we found that mothers of hybrid litters were slower to retrieve pups (see Results), we generated additional litters and recorded USVs on postnatal day 1 to determine whether reduced USV production in hybrid pups might contribute to slower maternal retrieval. Pups were removed individually from the home cage and placed in a cage with clean bedding inside the recording chamber, a $52 \times 36 \times 30$ cm anechoic foam-lined PVC box with a microphone (UltraSoundGate CM16/CMPA, Avisoft Bioacoustics) positioned approximately 15 cm above the floor of the box. Recording began immediately with vocalizations sampled at 192 kHz 16 bits using Avisoft-RECORDER software (v. 4.2.24) and hardware (UltraSoundGate 116Hb). Pups were recorded for 2 min during the light cycle (13.00–17.00 h). The number of vocalizations (distinct notes) produced/2 min was scored manually in Raven (v. 1.4).

## 2.7. Statistical analyses

All statistical analyses were conducted in R (v. 3.3.1). Open-field trials were analysed using generalized linear models (GLMs) with a Gaussian distribution (Shapiro–Wilk test: $p \geq 0.20$; lines crossed, time in centre, latency to enter the centre) or a negative binomial distribution (Shapiro–Wilk test: $p < 0.001$; time frozen at the start of trial) in the nlme package, with pup genotype, days to parturition and their interaction as explanatory variables. Pup retrieval results were analysed with repeated measures analysis of variance (ANOVA) using the nlme and car packages, with maternal ID and maternal experience included as random effects. *Post hoc* comparisons were conducted using least square means

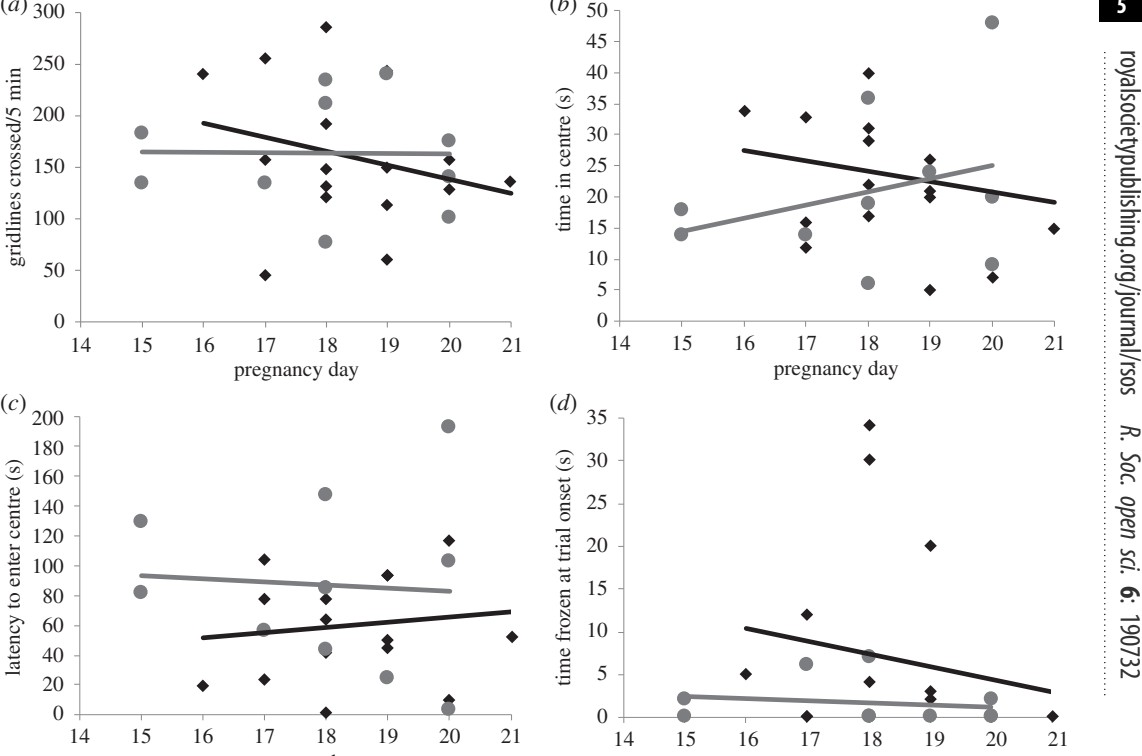

**Figure 1.** Activity (*a*) and anxiety-like behaviours (*b–d*) of females pregnant with hybrid (grey circles) or conspecific (black diamonds) litters in the open field test. (*a*) Activity measured as number of gridlines crossed, (*b*) time in the centre of the arena, (*c*) latency to enter the centre of the arena and (*d*) time frozen at the start of the test. Individual values are plotted by pregnancy day.

(LSM) in the lsmeans package with a Bonferroni correction. Home-cage activity was analysed using linear-mixed models (LMM), using the lme function in the nlme package. Pup genotype, day postpartum and their interaction were used as explanatory variables, with maternal ID and prior maternal experience as random effects. At time points where maternal activity levels differed qualitatively (non-overlapping standard errors), the effect of pup genotype was tested with *post hoc* ANOVAs. Suckling and milk let-down were analysed with repeated measures ANOVA using the nlme and car packages, with pup genotype, time and their interaction as explanatory variables. Maternal ID was included as a random effect. USVs were analysed with LMM using the lme function in the nlme package, including litter ID as a random effect. Model selection for open-field trials, USV production and home-cage activity was conducted using Akaike Information Criterion (corrected for small sample sizes) where the model with the lowest ΔAICc value was chosen as the best representative model for the data (AICc tables provided in electronic supplementary material, file S1). R code for statistical analyses is provided in the supplemental material (electronic supplementary material, file S2).

# 3. Results

## 3.1. No effect of litter genotype on anxiety-like behaviour in near-term females

There was no significant difference between females carrying hybrid litters ($n = 10$) and females carrying conspecific litters ($n = 16$) for any behavioural measure in the open-field test (figure 1). Using GLMs, all behavioural measures were best explained by the null model (electronic supplementary material, file S1), indicating no effect of litter genotype or pregnancy day. Using the criterion of Burnham & Anderson [45], we also considered the second-best models with ΔAICc less than 2. The second-best model for number of lines crossed (ΔAICc = 1.9) included pregnancy day, whereas second-best models for latency to enter the centre (ΔAICc = 0.3) and time frozen at start (ΔAICc = 0.5) included pup genotype. However, these effects were not significant in the models (electronic supplementary material, file S1). Most metrics for

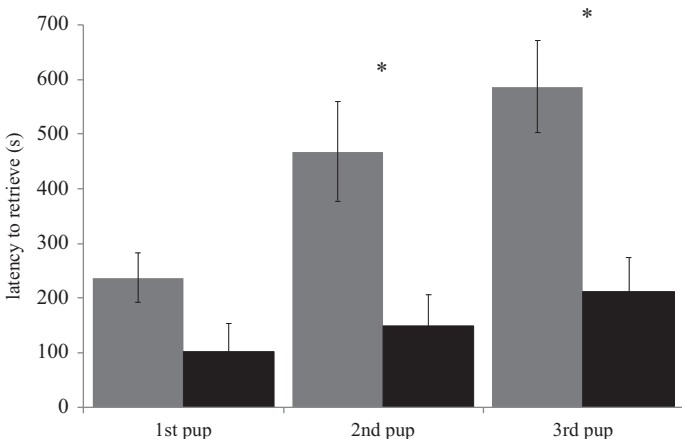

**Figure 2.** Maternal retrieval of hybrid (grey) or conspecific (black) pups. Females with hybrid offspring took longer to retrieve each pup compared to females with conspecific offspring. Bars represent mean ± 1 s.e. *$p_{Bonferroni-adjusted} < 0.05$ (LSM).

the open-field test were weakly correlated (|Pearson's $r$| less than 0.19). The number of lines crossed and latency to enter the centre of the arena were moderately negatively correlated (Pearson's $r = -0.58$).

## 3.2. Mothers of hybrid offspring are slower to retrieve pups

Females with hybrid offspring ($n = 12$) took significantly longer overall to retrieve pups than females with conspecific offspring ($n = 13$) (repeated measures ANOVA: $F_{1,46} = 7.62$, $p = 0.0014$), and significantly longer to retrieve second and third pups (LSM: $p_{Bonferroni-adjusted}$ less than 0.05) (figure 2). Prior maternal experience was not included in the best model for the repeated measures ANOVA. We recorded day 1 USVs from six hybrid and two conspecific litters. While the best model for the number of USVs produced included pup genotype as a fixed effect, genotype was not significant within the model (LMM: value ± s.e.: $32.55 ± 22.89$, $t = 1.42$, $p = 0.20$, electronic supplementary material, figure S1). However, when pups from the same litter were treated as independent samples, hybrids vocalized significantly more than conspecifics (ANOVA: $F_{1,32} = 5.34$, $p = 0.027$), suggesting that inclusion of more litters would reveal an effect of genotype in the full model. Importantly, there was no evidence that hybrid pups produced fewer USVs than conspecific pups.

## 3.3. Mothers of hybrids spend less time in nest on the first night postpartum

Home-cage activity was split into light and dark 12 h cycles, and the cycles were analysed separately. In the dark cycle, females with hybrid offspring ($n = 13$) were significantly more active and therefore spent less time in the nest on the first night postpartum relative to females with conspecific offspring ($n = 15$, ANOVA: $F_{1,28} = 4.95$, $p = 0.03$; figure 3). The nocturnal activity of mothers of hybrids decreased significantly thereafter (LMM: value ± s.e.: $-5.07 ± 2.08$, $t = -2.44$, $p = 0.017$) and was indistinguishable from that of females with conspecific offspring by the second night. These effects were explained by pup genotype alone, with prior maternal experience not included in the best model. Females with hybrid offspring spent more time in the nest during the light cycle on the fourth day postpartum (ANOVA: $F_{1,28} = 7.33$, $p = 0.011$, figure 3). However, there was no overall effect of pup genotype on maternal home-cage activity in the light cycle (LMM: value ± s.e.: $-1.64 ± 1.14$, $t = -1.44$, $p = 0.15$), nor was there an effect of day postpartum (value ± s.e.: $0.023 ± 0.78$, $t = 0.029$, $p = 0.97$).

## 3.4. No effect of offspring genotype on suckling and milk let-down

In the milk let-down assay, the change in mass of females with hybrid offspring ($n = 14$) did not differ from that of females with conspecific offspring ($n = 15$) (repeated measures ANOVA: $F_{1,25} = 0.017$, $p = 0.89$; figure 4a). Similarly, hybrid ($n = 57$) and conspecific ($n = 51$) pups did not differ in suckling efficiency based on the overall change in mass (repeated measures ANOVA: $F_{1,25} = 0.082$, $p = 0.78$; figure 4b). However, hybrids lost less weight during the 2 h separation from mothers (ANOVA: $F_{1,72} = 4.27$, $p = 0.042$).

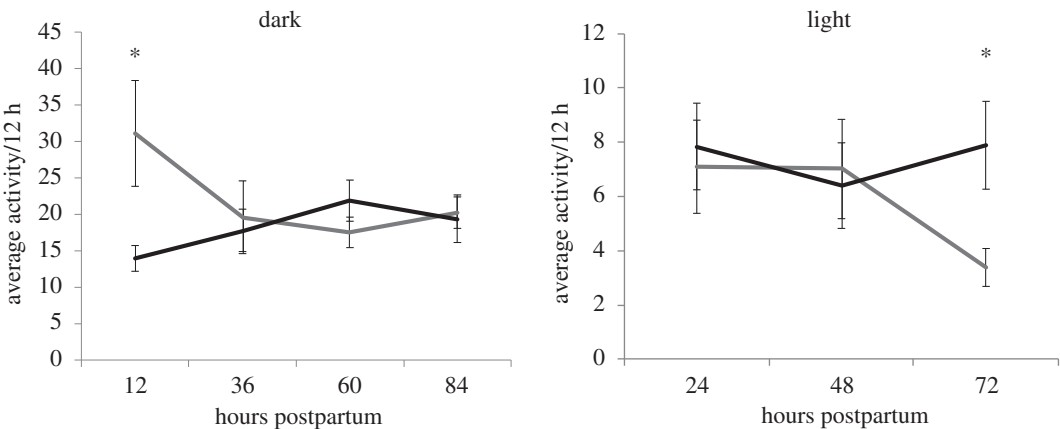

**Figure 3.** Maternal home-cage activity in the dark cycle and the light cycle. Females with hybrid offspring (grey) were out of the nest more on postnatal day 1 in the dark cycle but were more similar to females with conspecific offspring (black) over time. Females were similar in the light cycle until day 4, when females with hybrid offspring spent more time in the nest. Error bars represent ± 1 s.e. *$p < 0.05$ (ANOVA).

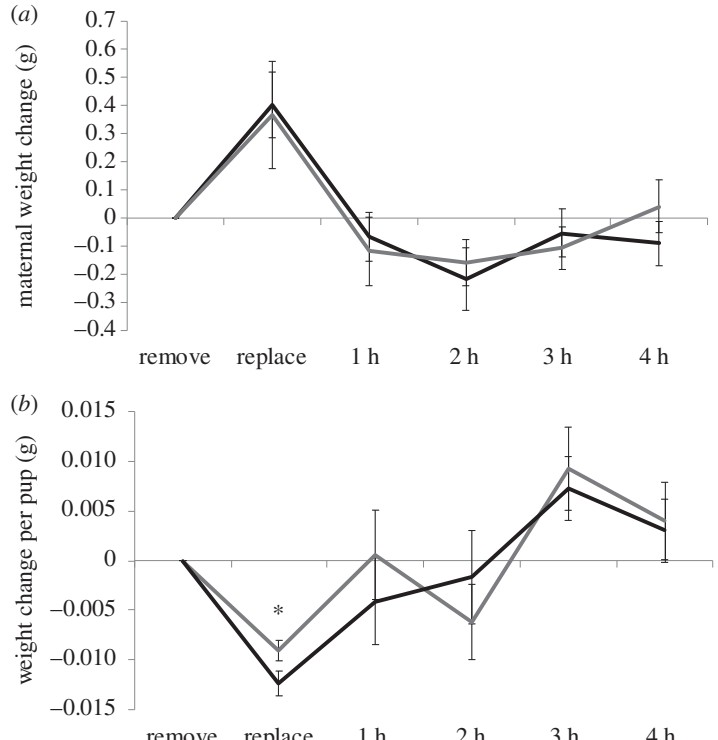

**Figure 4.** Maternal milk let-down (*a*) and pup suckling efficiency (*b*) for hybrid (grey) and conspecific (black) litters. Pup genotype did not affect maternal milk let-down. Pup weight changes only differed when separated from mothers, during which time hybrids lost less weight than conspecific pups. Error bars represent ± 1 s.e. *$p < 0.05$ (ANOVA).

## 4. Discussion

We tested for behavioural effects of exposure to abnormal placental signals in *M. m. domesticus* mothers of hybrid relative to conspecific litters. Females pregnant with hybrid offspring did not differ in anxiety-like behaviours in the open-field test but were slower to retrieve newborn pups and spent less time in the nest on the first night after pups were born compared to females with conspecific offspring. After this initial deficit in response to pups, mothers of hybrids did not differ from mothers of conspecifics in time spent in the nest, and there was no evidence for a physiological effect of pup genotype on milk let-down on postpartum day 5. Taken together, these results suggest that placental dysregulation leads to poor

maternal priming in the mothers of hybrid offspring, the effect of which is alleviated by continued exposure to pups. We discuss these findings in relation to the mechanistic basis of placental effects on mothers, and the role of imprinted genes.

## 4.1. Placental effects on prepartum maternal anxiety

The direction and strength of the relationship between maternal anxiety and maternal care differ between taxa and experimental designs, with higher anxiety associated with impaired maternal care in some human and non-human primate studies [46–48] but variable effects of anxiety in rodent mothers [49,50]. Whereas pup-directed behaviour was initially impaired in mothers of hybrids (discussed below), we did not find evidence for effects of litter genotype on anxiety-like behaviours in near-term mothers. We note that sample size for mothers carrying hybrids in this assay was small ($n = 10$) and that measuring behaviour during the last week of gestation rather than on a single day introduced additional variance within the two groups of females. We cannot, therefore, rule out the possibility that an effect of pup genotype might be detected with larger sample sizes. Indeed, the inclusion of pup genotype in second-best models for two measures of anxiety-like behaviour (latency to enter centre and time frozen at start) suggests that this might be the case. Interestingly, our results are broadly similar to those obtained in a study of the effects of a single imprinted gene knockout, in which wild-type mothers of *Peg3* knockout litters did not exhibit altered anxiety-like behaviour prepartum but were slower to retrieve neonates [26].

## 4.2. Placental effects on postpartum maternal behaviours

The role of placental hormones in priming the onset of maternal care was demonstrated over 20 years ago in rats: placental lactogen infusion into the medial preoptic area (MPoA) of the hypothalamus accelerates maternal responsiveness to foster pups in steroid-primed, nulliparous females [20,51]. Recent work supports analogous action of placental lactogens in the maternal MPoA in mice [52]. In the natural hybrid system studied here, the expression of 14 placental lactogens, including the strongest candidate for maternal priming (*Prl3b1*, formerly PL-II) [53], is significantly reduced in hybrid relative to *M. m. domesticus* placentas [34]. Thus, females carrying hybrid litters are exposed to weaker placental signals than females carrying conspecific offspring. While our data do not demonstrate a causal link between altered placental signalling and early postpartum deficits in maternal responsiveness, both the direction and the transient nature of the differences between mothers of hybrids and conspecific pups are consistent with weaker maternal priming in hybrid pregnancies.

Virgin female mice require only three 2 h exposures to neonatal foster pups to induce a full suite of pup-directed maternal behaviours, including retrieval, licking and grooming and nursing posture [54]. Thus, sensory stimuli from newborn pups are sufficient to activate the neural circuitry subserving maternal care, even in the absence of placental priming or the maternal hormonal changes associated with parturition and lactation [55]. In our study, females with hybrid litters underwent normal parturition and we found no evidence for deficits in lactation. There were no qualitative differences between hybrid and conspecific litters in the presence of milk bands (visible indicators of milk intake) at the time of the pup retrieval test, and no quantitative differences in maternal milk let-down on day 5. Collectively, these observations suggest that weaker signals from hybrid placentas cause deficits in the onset of maternal behaviour without major impact on maternal physiology, while normalization of maternal behaviours by approximately 24 h postpartum is a consequence of continuous exposure to pups. Given that the MPoA is both a major target of placental lactogens in late pregnancy [20,56], and a key mediator of appetitive maternal behaviours (e.g. pup retrieval) postpartum [57], we speculate that maternal perception of newborn pups as rewarding [58,59] is impaired in poorly primed mothers of hybrids, whereas somatosensory stimuli from the pups themselves raise maternal motivation to normal levels, with a delay comparable to that seen in virgin females.

## 4.3. Contribution of imprinted genes to placental effects on mothers

The placenta is both the site of resource transfer from mother to offspring and the source of hormonal signals that target the maternal brain. Correct dosage of placental imprinted genes is critical to both functions [4,10,15,17]. However, while placental misexpression of individual imprinted genes affects conceptus growth in predictable and repeatable directions [60,61], effects on maternal behaviour and physiology are considerably less studied and can differ across studies [25,27]. To our knowledge, the

effects of altered placental expression on wild-type female mice have been tested for only three imprinted genes: *Peg3*, *Grb10* and *Phlda2* [25,26,28,30]. In the hybrid system studied here, *Grb10* and *Phlda2* are among the 13 imprinted genes with significantly altered expression in hybrid relative to *M. m. domesticus* placenta [34]. Specifically, *Grb10* is upregulated in hybrid placenta relative to *M. m. domesticus* and *Phlda2* is overexpressed relative to both parental species [34]. Whereas placental *Grb10* influences maternal metabolism and nutrient transfer [28], phenotypes not measured here, *Phlda2* negatively regulates placental endocrine cell lineages and has distinct effects on postnatal maternal behaviour up to 4 days postpartum when over- versus under-expressed [29,30]. Despite many differences in study system and experimental design between the work presented here and that of Creeth and colleagues on *Phlda2* [30], we find broadly concordant negative effects on maternal pup-directed behaviours in association with overexpression of placental *Phlda2*. Thus, the behavioural results of the present study, together with our prior work on hybrid placental gene expression [34], provide independent support for the proposition that placental *Phlda2* is an important effector of the onset of maternal care in mice [30].

We note that other imprinted genes, together with non-imprinted genes, are misexpressed in hybrid placenta [34]. It is therefore unlikely that *Phlda2* overexpression is the sole cause of altered behaviour in mothers of hybrids. Likewise, the fact that maternal behaviours are affected by manipulation of either *Peg3* or *Phlda2* in offspring [26,30] indicates that neither gene is sufficient for induction of 'normal' maternal behaviour and suggests that targeted manipulation of other imprinted genes will uncover additional placental effects on mothers. Future work on the *M. m. domesticus*/*M. spretus* cross will use backcross mapping to identify genomic intervals that influence maternal behaviour. This is a promising strategy for assessing the contribution of imprinted genes, as most are clustered in the genome and two such clusters contain multiple genes with altered expression in hybrid placenta [34].

# 5. Conclusion

The indelible effects of maternal physiology and behaviour on offspring phenotypes are well established across a range of viviparous and oviparous taxa [62–65]. The endocrine function of the mammalian placenta provides a unique mechanism for reciprocal effects of unborn offspring on the physiology and behaviour of pregnant and nursing females [10,20]. While these placental effects on adult females are transient relative to maternal effects on developing offspring, the impact of dysregulated placental signalling on mothers, and therefore on offspring, is significant. For example, reduced placental *Peg3* expression in human pregnancies is associated with maternal prenatal depression [66], while prenatal depression is a predictor of lower growth rate and higher disease risk in infants [67]. Here, we show that female house mice exposed to placental dysregulation are slower to retrieve neonates displaced from the nest and spend less time in the nest with neonates relative to control mothers. In nature, the potential fitness costs of these deficits in maternal behaviour are high: both slow retrieval and reduced nest attendance could increase the probability of predation and hypothermia in altricial neonates. The results of this study motivate further work on the mechanistic basis of placental effects on mothers, and the contribution of placental function and dysfunction to maternal effects on offspring.

Ethics. All animal procedures were approved by the Oklahoma State University IACUC under protocol # AS-1-41. Thirty-three adult female *M. m. domesticus* (mean age ± s.d.: 116 ± 54 days) were paired to either a male *M. m. domesticus* or male *M. spretus* (163 ± 82 days, *n* = 33) for 14 days. *M. m. domesticus* was represented by the WSB/EiJ strain (Jackson Laboratory) and *M. spretus* was represented by the SFM/Pas strain (Montpellier Wild Mice Genetic Repository). The wild-derived inbred mice used in this study were maintained on a 12 L : 12 D cycle with lights on at 09.30 and were provided with ad libitum food (LabDiet® 5001 Rodent Diet) and water. Adult males and females were housed singly in polycarbonate cages with SaniChips® unless paired for mating. After the experiment, males and females were used as breeders in the main colony.
Data accessibility. The dataset and code have been uploaded as supplementary files.
Authors' contributions. S.G. conducted data collection and analysis, participated in the design of the study and drafted the manuscript; P.C. conceived of the study and critically revised the manuscript; J.L.G. provided feedback on study design. All authors gave final approval for publication.
Competing interests. We declare no competing interests.
Funding. The work was funded by NSF award IOS-1558109 to P.C., and an Animal Behavior Society Student Research grant and a Society for Integrative and Comparative Biology Grant-in-Aid of Research to S.G.
Acknowledgements. The manuscript was improved by the thoughtful comments of two anonymous reviewers. We thank members of the Campbell and Grindstaff labs for their support and feedback throughout this project and Lena Arévalo

for input on the manuscript. We thank the Clarke lab at Oklahoma State University for extended loan of activity monitoring equipment and the Good lab at University of Montana for the *M. spretus* strain used in this study. P.C. thanks Katya Mack for generously sharing the ideas that catalysed the project.

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
