## [Reviewer comments · Royal Society Open Science]

Review History

RSOS-190732.R0 (Original submission)

Review form: Reviewer 1

Is the manuscript scientifically sound in its present form?

Yes

Are the interpretations and conclusions justified by the results?

No

Is the language acceptable?

Yes

Is it clear how to access all supporting data?

Yes

Do you have any ethical concerns with this paper?

No

Have you any concerns about statistical analyses in this paper?

Yes

Recommendation?

Major revision is needed (please make suggestions in comments)

Comments to the Author(s)

This study examined the effects of altered placental signaling on maternal behavior. It compared the behavior of females who had hybrid offspring, with concomitant reduced placental size and differential expression of placental endocrine genes (e.g., lactogens), versus those that had conspecific offspring. They found evidence that the placenta is critical in priming maternal behavior, with mothers of hybrid offspring having reduced maternal responsiveness. There were ambiguous results regarding maternal anxiety – the direction of the result depended on the measure. These differences largely disappeared over time, suggesting that exposure to pups mitigated differences in placental signaling.

I found the paper very well written, interesting, and a joy to read, especially the Introduction and Discussion. My major criticisms are 1) a lack of detail regarding the statistical tests (e.g., Why were particular tests used? Which packages/functions in R were used? How were model assumptions evaluated?), and 2) some quibbles with how interaction coefficients are interpreted. Unless the results are strongly affected by addressing these issues (which the figures suggest probably won't be the case), then it will be great to see this study published! I really must emphasize, however, that it is impossible to fully understand how the data was analyzed, or why it was analyzed that way, from what is written.

I give my specific comments below:

Introduction

Lines 72-73: “imprinted genes are predominantly expressed from only one allele”. I personally found this wording a little awkward; perhaps “only a single allele of an imprinted gene is predominantly expressed”?

Lines 79-80: “placental endocrine compartment”. I can more-or-less figure out what this means through context, but later in the text (Lines 117-118), an actual definition is given, and it would be helpful if that definition was moved up to here.

Line 85: “Females with silenced Peg3”. Does this mean neither the paternal nor the maternal allele is expressed? Please clarify.

Lines 82-100: Somewhat related, I was curious whether there have been any experiments expressing the normally non-expressed allele?

Line 102-105: I was curious about these phrases: “natural system” and “misexpressed”. What makes this a more “natural system” than other mouse work? Does a hybrid like this ever occur in the wild? These are two strains that were lab created and then crossed, correct? It's a bit of a quibble, but I think there might be something meant here that isn't being fully expressed, and if so, it would be great to have it explicated more clearly. Maybe the real distinction here is between single gene vs. multiple gene manipulations? Related to this, I was curious about the use of “misexpressed”. Are these genes expressed the same way in both conspecific crosses? Is it the regulation of expression and/or allele silencing that is disrupted by the hybrid cross?

Methods

Lines 158-161: Was this variation in experience ever utilized in statistical analyses?

Lines 177-178: Clarify that anxiety is expressed by not going into the center (or staying near the edge).

Line 191: Clarify that all pups were returned to the nest after the test . . .

Lines 193-198: Can you give some additional information about why this schedule was chosen? Does more suckling happen at night?

Line 200: Can you be more specific about what “activity levels” means? Is it the distance traveled?

Lines 218-229: I found that this section was sorely lacking a lot of critical details. None of the packages and functions used were listed (especially for LMMS, this is a problem because you can get different results depending on the package and function used for p-value estimation). Why ANOVAs were used in some cases, but mixed models in others was also not explained. Why at some point a general linear model was used (and what error structure/link function was used) was not described. Effects are sometimes referred to as “fixed” and other times as “explanatory”. Were model assumptions ever checked? The role AIC played in analysis wasn’t fully explained. If an analysis is done post-hoc to further examine a main or interaction effect, this should be clearly stated. Finally, it would be very helpful to present the methods for each assay in the same order as the analyses and likewise, in the same order for the Results. This paragraph for me really sunk an otherwise good paper.

Line 220: It took me some time to realize that “time point” here referred to the first, second, third pup. I find this analysis completely inappropriate as the three times are completely correlated. The time to get the second pup is the time to get the first pup plus the time to get the second pup, for example. Certainly, Figure 2 shows an extremely linear relationship.

Line 225-226: The interaction was also included according to the Results.

Results

Line 236: Any reason for such a different sample size from the conspecific cross? Are hybrid litters harder to produce?

Line 241: The figure makes it appear that hybrid females remain unchanged over time, and it is only the conspecific females that are driving the main effect of activity being reduced towards parturition.

Line 244-245: Isn't it more accurate to interpret this positive coefficient as the two lines are becoming more different?

Line 248-249: Again, I think the correct interpretation of the (here negative) interaction coefficient is that the lines become more similar.

Line 250: “In contrast . . .” I felt that these conflicting results were not explored very much in the Discussion. They were summarized well enough, but I wonder if the authors could speculate more on why there might be this difference.

Line 262: I’m repeating myself, but what is the justification for analyzing each pup retrieval as independent events?

Lines 267-269: Given this sentence, I'm curious how "important" the random effect is. There are a number of different ways "important" could be quantified . . .

Lines 277-288: What are these ANOVAs? Are they some kind of post-hoc comparison of specific time points?

Line 298: Is this a main effect of offspring genotype?

Line 299 and 300: Specify that these are ANOVAs(?)

Review form: Reviewer 2

Is the manuscript scientifically sound in its present form?

No

Are the interpretations and conclusions justified by the results?

No

Is the language acceptable?

Yes

Is it clear how to access all supporting data?

Yes

Do you have any ethical concerns with this paper?

No

Have you any concerns about statistical analyses in this paper?

Yes

Recommendation?

Accept with minor revision (please list in comments)

Comments to the Author(s)

The authors present their study on the impact of hybrid offspring on an exposed dams' behaviour pre and postnatally. They report alterations in anxiety and also pup directed behaviour which, in combination with the bioRxiv submission (Arévalo, L., and P. Campbell. 2019. Placental effects on maternal brain revealed by disrupted placental gene expression in mouse hybrids) lends further support to the hypothesis that signals from the placenta plays a key role in the induction of aspects of maternal care, and that disruption of imprinting in the placenta can lead to deficits in maternal care via disruption of these signals. The study is elegantly designed and the results are important.

I have some points regarding the litters used in the analysis that require clarification

The authors mention that "Females used in this study were either first (26/41 litters), second (11/41 litters), or third (2/41 litters) time mothers". They also mention that "Litters were only used in this behavioral test if the number of pups in the litter was three or more". Both parity and litter size are known to be factors influencing maternal behaviour.

How many litters were used in the analysis with less than three pups? If data from these is removed, are the differences detected still significant?

Were there differences in any test in relation to litter size within group? The authors should do a correlation analysis.

Were there differences in any test between first, second and third time dams within group? Anova would be a useful test.

Was the composition of first, second and third time dams balanced statistically across the groups? Also, the number of litters does not add up to 41?

Minor

Line 25

“how altered placental signaling might affect maternal behavior is unstudied in a natural system”
Could the author rephrase this sentence as this is not the case (Creeth et al 2018)

Line 96

“with presumed secondary effects on placental hormone production (Tunster et al. 2016)”.
Not presumed – demonstrated in earlier papers

Line 97

“Notably, suppression of placental Phlda2 expression in wild-type females increases maternal nursing and grooming of newborn pups, while over expression increases non-pup-directed maternal behaviors such as nest building (Creeth et al. 2018)”.
This sentence needs to be rephrased to be clearer. Creeth et al. reported that loss of function of Phlda2 in the fetally-derived placenta resulted in increased maternal nurturing by wild type dams whereas loss-of-imprinting of Phlda2 (increased expression) resulted in an increased focus of wild type dams on nest building.

Line 101

After the first sentence – include the reviews by Creeth et al 2019 and Potter et al published in Front Neuroendocrinol. 2019 which extensively cover this topic

Line 393

“Likewise, the fact that maternal behaviors are affected by manipulation of either Peg3 or Phlda2 in offspring (Curley et al. 2004; Creeth et al. 2018)”
Curley et al. 2004 reports on loss of function of Peg3 in the dam, not the offspring. McNamara 2018 reports on loss of function of Peg3 in the offspring

Decision letter (RSOS-190732.R0)

25-Jun-2019

Dear Ms Gardner

On behalf of the Editors, I am pleased to inform you that your Manuscript RSOS-190732 entitled "Placental genotype affects pre- and early postpartum maternal behaviour" has been accepted for publication in Royal Society Open Science subject to minor revision in accordance with the referee suggestions. Please find the referees' comments at the end of this email.

The reviewers and handling editors have recommended publication, but also suggest some minor

revisions to your manuscript. Therefore, I invite you to respond to the comments and revise your manuscript.

- Ethics statement

- Data accessibility

If you wish to submit your supporting data or code to Dryad (<http://datadryad.org/>), or modify your current submission to dryad, please use the following link:
<http://datadryad.org/submit?journalID=RSOS&manu=RSOS-190732>

- Competing interests

- Authors' contributions

- Acknowledgements

- Funding statement

Please ensure you have prepared your revision in accordance with the guidance at

<https://royalsociety.org/journals/authors/author-guidelines/> -- please note that we cannot publish your manuscript without the end statements. We have included a screenshot example of the end statements for reference. If you feel that a given heading is not relevant to your paper, please nevertheless include the heading and explicitly state that it is not relevant to your work.

Because the schedule for publication is very tight, it is a condition of publication that you submit the revised version of your manuscript before 04-Jul-2019. Please note that the revision deadline will expire at 00.00am on this date. If you do not think you will be able to meet this date please let me know immediately.

Please note that Royal Society Open Science charge article processing charges for all new

submissions that are accepted for publication. Charges will also apply to papers transferred to Royal Society Open Science from other Royal Society Publishing journals, as well as papers submitted as part of our collaboration with the Royal Society of Chemistry (<http://rsos.royalsocietypublishing.org/chemistry>).

on behalf of Dr Alexander Ophir (Associate Editor) and Kevin Padian (Subject Editor)
openscience@royalsociety.org

Associate Editor Comments to Author (Dr Alexander Ophir):

Dear Dr. Gardner,

I have now received reviews from two experts in the field who were both impressed by your study. Each reviewer raised some points that you will need to address before publication, most notably justifying and clarifying your statistical approaches and interpretations from them. I believe addressing these and the other minor points details below should be relatively simple to address. I agree with both of the reviewers that you have conducted an elegant and interesting and well written study that will be an important addition to the literature once these reviewers comments are addressed.

Best
Alex Ophir
Associate Editor RSOS

Reviewer comments to Author:
Reviewer: 1

Comments to the Author(s)

This study examined the effects of altered placental signaling on maternal behavior. It compared the behavior of females who had hybrid offspring, with concomitant reduced placental size and differential expression of placental endocrine genes (e.g., lactogens), versus those that had conspecific offspring. They found evidence that the placenta is critical in priming maternal behavior, with mothers of hybrid offspring having reduced maternal responsiveness. There were ambiguous results regarding maternal anxiety – the direction of the result depended on the

measure. These differences largely disappeared over time, suggesting that exposure to pups mitigated differences in placental signaling.

I found the paper very well written, interesting, and a joy to read, especially the Introduction and Discussion. My major criticisms are 1) a lack of detail regarding the statistical tests (e.g., Why were particular tests used? Which packages/functions in R were used? How were model assumptions evaluated?), and 2) some quibbles with how interaction coefficients are interpreted. Unless the results are strongly affected by addressing these issues (which the figures suggest probably won't be the case), then it will be great to see this study published! I really must emphasize, however, that it is impossible to fully understand how the data was analyzed, or why it was analyzed that way, from what is written.

I give my specific comments below:

Introduction

Lines 72-73: "imprinted genes are predominantly expressed from only one allele". I personally found this wording a little awkward; perhaps "only a single allele of an imprinted gene is predominantly expressed"?

Lines 79-80: "placental endocrine compartment". I can more-or-less figure out what this means through context, but later in the text (Lines 117-118), an actual definition is given, and it would be helpful if that definition was moved up to here.

Line 85: "Females with silenced Peg3". Does this mean neither the paternal nor the maternal allele is expressed? Please clarify.

Lines 82-100: Somewhat related, I was curious whether there have been any experiments expressing the normally non-expressed allele?

Line 102-105: I was curious about these phrases: "natural system" and "misexpressed". What makes this a more "natural system" than other mouse work? Does a hybrid like this ever occur in the wild? These are two strains that were lab created and then crossed, correct? It's a bit of a quibble, but I think there might be something meant here that isn't being fully expressed, and if so, it would be great to have it explicated more clearly. Maybe the real distinction here is between single gene vs. multiple gene manipulations? Related to this, I was curious about the use of "misexpressed". Are these genes expressed the same way in both conspecific crosses? Is it the regulation of expression and/or allele silencing that is disrupted by the hybrid cross?

Methods

Lines 158-161: Was this variation in experience ever utilized in statistical analyses?

Lines 177-178: Clarify that anxiety is expressed by not going into the center (or staying near the edge).

Line 191: Clarify that all pups were returned to the nest after the test . . .

Lines 193-198: Can you give some additional information about why this schedule was chosen? Does more suckling happen at night?

Line 200: Can you be more specific about what "activity levels" means? Is it the distance traveled?

Lines 218-229: I found that this section was sorely lacking a lot of critical details. None of the packages and functions used were listed (especially for LMMs, this is a problem because you can get different results depending on the package and function used for p-value estimation). Why ANOVAs were used in some cases, but mixed models in others was also not explained. Why at some point a general linear model was used (and what error structure/link function was used) was not described. Effects are sometimes referred to as “fixed” and other times as “explanatory”. Were model assumptions ever checked? The role AIC played in analysis wasn’t fully explained. If an analysis is done post-hoc to further examine a main or interaction effect, this should be clearly stated. Finally, it would be very helpful to present the methods for each assay in the same order as the analyses and likewise, in the same order for the Results. This paragraph for me really sunk an otherwise good paper.

Line 220: It took me some time to realize that “time point” here referred to the first, second, third pup. I find this analysis completely inappropriate as the three times are completely correlated. The time to get the second pup is the time to get the first pup plus the time to get the second pup, for example. Certainly, Figure 2 shows an extremely linear relationship.

Line 225-226: The interaction was also included according to the Results.

Results

Line 236: Any reason for such a different sample size from the conspecific cross? Are hybrid litters harder to produce?

Line 241: The figure makes it appear that hybrid females remain unchanged over time, and it is only the conspecific females that are driving the main effect of activity being reduced towards parturition.

Line 244-245: Isn't it more accurate to interpret this positive coefficient as the two lines are becoming more different?

Line 248-249: Again, I think the correct interpretation of the (here negative) interaction coefficient is that the lines become more similar.

Line 250: “In contrast . . .” I felt that these conflicting results were not explored very much in the Discussion. They were summarized well enough, but I wonder if the authors could speculate more on why there might be this difference.

Line 262: I’m repeating myself, but what is the justification for analyzing each pup retrieval as independent events?

Lines 267-269: Given this sentence, I’m curious how “important” the random effect is. There are a number of different ways “important” could be quantified . . .

Lines 277-288: What are these ANOVAs? Are they some kind of post-hoc comparison of specific time points?

Line 298: Is this a main effect of offspring genotype?

Line 299 and 300: Specify that these are ANOVAs(?)

Reviewer: 2

Comments to the Author(s)

The authors present their study on the impact of hybrid offspring on an exposed dams' behaviour pre and postnatally. They report alterations in anxiety and also pup directed behaviour which, in combination with the bioRxiv submission (Arévalo, L., and P. Campbell. 2019. Placental effects on maternal brain revealed by disrupted placental gene expression in mouse hybrids) lends further support to the hypothesis that signals from the placenta plays a key role in the induction of aspects of maternal care, and that disruption of imprinting in the placenta can lead to deficits in maternal care via disruption of these signals. The study is elegantly designed and the results are important.

I have some points regarding the litters used in the analysis that require clarification

The authors mention that "Females used in this study were either first (26/41 litters), second (11/41 litters), or third (2/41 litters) time mothers". They also mention that "Litters were only used in this behavioral test if the number of pups in the litter was three or more". Both parity and litter size are known to be factors influencing maternal behaviour.

How many litters were used in the analysis with less than three pups? If data from these is removed, are the differences detected still significant?

Were there differences in any test in relation to litter size within group? The authors should do a correlation analysis.

Were there differences in any test between first, second and third time dams within group?

Anova would be a useful test.

Was the composition of first, second and third time dams balanced statistically across the groups?

Also, the number of litters does not add up to 41?

Minor

Line 25

"how altered placental signaling might affect maternal behavior is unstudied in a natural system"
 Could the author rephrase this sentence as this is not the case (Creeth et al 2018)

Line 96

"with presumed secondary effects on placental hormone production (Tunster et al. 2016)".

Not presumed - demonstrated in earlier papers

Line 97

"Notably, suppression of placental Phlda2 expression in wild-type females increases maternal nursing and grooming of newborn pups, while over expression increases non-pup-directed maternal behaviors such as nest building (Creeth et al. 2018)".

This sentence needs to be rephrased to be clearer. Creeth et al. reported that loss of function of Phlda2 in the fetally-derived placenta resulted in increased maternal nurturing by wild type dams whereas loss-of-imprinting of Phlda2 (increased expression) resulted in an increased focus of wild type dams on nest building.

Line 101

After the first sentence - include the reviews by Creeth et al 2019 and Potter et al published in Front Neuroendocrinol. 2019 which extensively cover this topic

Line 393

“Likewise, the fact that maternal behaviors are affected by manipulation of either Peg3 or Phlda2 in offspring (Curley et al. 2004; Creeth et al. 2018)”
 Curley et al. 2004 reports on loss of function of Peg3 in the dam, not the offspring. McNamara 2018 reports on loss of function of Peg3 in the offspring

Author's Response to Decision Letter for (RSOS-190732.R0)

See Appendix A.

RSOS-190732.R1 (Revision)

Review form: Reviewer 1

Is the manuscript scientifically sound in its present form?

Yes

Are the interpretations and conclusions justified by the results?

Yes

Is the language acceptable?

Yes

Do you have any ethical concerns with this paper?

No

Have you any concerns about statistical analyses in this paper?

Yes

Recommendation?

Major revision is needed (please make suggestions in comments)

Comments to the Author(s)

This well-written paper describes a well-conceived study that examined the effects of altered placental signaling on maternal behavior. It compared the behavior of females who had hybrid offspring versus those that had conspecific offspring. The Authors found indirect evidence that the placenta (which previous work showed was reduced when offspring were hybrid) is critical in priming maternal behavior, with mothers of hybrid offspring having reduced maternal responsiveness and being more anxious (at least earlier in pregnancy). The reduction in genotype effect over time suggested that exposure to pups mitigated the effects generated by differences in placental signaling.

I very much appreciate the changes the Authors made in response to my and the other Reviewer's comments. I have detailed comments below, but my major criticisms are still about the description of the statistical analyses. I understand the authors made some attempt to address my previous criticisms of their statistical analyses section. However, I still have strong concerns.

In particular, 1) what is said in the text often does not match what is in the included R code, 2) it is not specifically stated in the text whether model assumptions were examined (in particular, I'd like to know whether their data were overdispersed in models where they assumed a Poisson distribution), and 3) LMMs are not consistently analyzed (two different R packages are used, lme4 and nlme, with no explanation of why the switch). These discrepancies make me uncomfortable with the manuscript and I hope they can be addressed.

More specific comments:

Introduction:

Line 77: Would it be worth specifying "maternal" hypothalamus to make it clearer? At least that is my inference; if I am wrong about this, then I STRONGLY advise adding a clarifying descriptor.

Methods:

Lines 160-161: "starting when the mouse first entered the outer edge of the grid." Can you explain? Did they generally go to the edge first and then later move into the center (I know from below that some of them simply froze...)?

Lines 206: Could you include in the Results how the variables from the open field assay are correlated?

Line 207: Many of your GLMs used a Poisson distribution. Did you ever verify this was appropriate? For example, state that you examined the residuals for normality in the Methods (which you said you did in the response to my previous comments). Also, did you test to see if your data were overdispersed? If so, a Poisson distribution would be inappropriate, although perhaps using a the "quasi-poisson" family would be OK.

Line 207: "the lme4 package" This statement is incorrect given the code you shared (which uses the "glm" function) and the fact that functions in lme4 require a random statement, and you don't have one here.

Lines 209-210: "for pairwise comparisons at each time point" I don't understand what this means. That's not what a repeated-measures ANOVA does if you are modeling time as a continuous variable (if you are not, you should defend this).

Line 210: "using the car package" Something else must have been used to make this model because no function in the car package does this. My guess is that you used car to generate ANOVA tables.

Line 210-211: "with maternal ID and maternal experience included as random effects." Your R code only has maternal ID in the random statement.

Line 213: "the lme function" Why use lme4 for one set of models and nlme for the other?

Line 215-216: "At time points where maternal activity levels differed" Differed how? How was this determined?

Line 217: "repeated measures ANOVA" Using what function/package?

Line 217: "time" Not in the R code, although it should have been; instead it is included nested within maternal ID in the random statement. It's not clear why this was done here and not in your other repeated-measures ANOVAs.

Figure 1.

I still have some reservations about how the results from the open field assay were interpreted, although I very much appreciate how the authors incorporated my feedback on their previous version. I would re-word the caption to follow the description given in the beginning: "Activity and anxiety-like behaviors". For example, (a) says "Activity", whereas (b-d) are presumably "anxiety-like" behaviors? The main comment I have is that I'm a little skeptical of making a big deal about main effects in (a) and (b) when they are not apparent in the figure and there was a significant interaction effect. Because of the significant interaction, it is clearly not the case that one genotype was always higher than the other.

Discussion:

Lines 305-306: “females carrying hybrid litters spent more time in the center of the arena as pregnancy progressed”. What is the earliest day at which the hybrid placenta could be impacting the maternal brain? Because it seems that in all these cases, the hybrid and conspecific moms start out different, but become similar closer to parturition. This result in particular seems to not fit this statement. By parturition, the two genotypes did not appear to differ at all. Rather earlier in pregnancy, mothers with hybrid litters seemed more anxious.

Line 347: “Grb10 and Phlda2” Can you be explicit about the direction of differential expression, rather than just that there are differences? For example, much later in this paragraph it is said that Phlda2 is overexpressed in hybrids. It would be nice to know that sooner.

Line 364: “will focus on isolation of genomic intervals” This is foreshadowing some kind of GWAS or QTL study?

Review form: Reviewer 2

Is the manuscript scientifically sound in its present form?

Yes

Are the interpretations and conclusions justified by the results?

No

Is the language acceptable?

Yes

Do you have any ethical concerns with this paper?

No

Have you any concerns about statistical analyses in this paper?

Yes

Recommendation?

Accept with minor revision (please list in comments)

Comments to the Author(s)

The authors have responded to the majority of comments made on their original submission. However, in our original review, we asked “How many litters were used in the analysis with less than three pups? If data from these is removed, are the differences detected still significant?” The authors have not directly answered this question. Instead, the authors restate that there is no difference in litter size between the 24 conspecific and 17 hybrid litters. However, all 41 litters are not used in every test ex. prepregnancy activity N = 16 and N = 10; retrieval N = 13 and N = 12; post pregnancy activity N = 15 and N = 13; milk let down N = 15 and N = 14 and USVs N = 3 and N = 6. It is important to demonstrate that parity and litter size are balanced for the animals included in each test, not those used overall.

“Likewise, the fact that maternal behaviors are affected by manipulation of either Peg3 or Phlda2 in offspring (Curley et al. 2004; Creeth et al. 2018)” implies that Curley et al., 2004 included a study on maternal behaviour in a model where WT dams carry and care for Peg3 mutant pups. They report on litter size, pup weight gain per se and after separation, and

maternal weight gain but did specifically examine maternal behaviour for WT dams that carry and care for Peg3 mutant pups. Therefore this citation is incorrect.

Decision letter (RSOS-190732.R1)

08-Aug-2019

Dear Ms Gardner,

On behalf of the Editors, I am pleased to inform you that your Manuscript RSOS-190732.R1 entitled "Placental genotype affects pre- and early postpartum maternal behaviour" has been accepted for publication in Royal Society Open Science subject to minor revision in accordance with the referee suggestions. Please find the referees' comments at the end of this email.

The reviewers and Subject Editor have recommended publication, but also suggest some minor revisions to your manuscript. Therefore, I invite you to respond to the comments and revise your manuscript.

- Ethics statement

- Data accessibility

<http://datadryad.org/submit?journalID=RSOS&manu=RSOS-190732.R1>

- Competing interests

- Authors' contributions

- Acknowledgements

- Funding statement

Because the schedule for publication is very tight, it is a condition of publication that you submit the revised version of your manuscript before 17-Aug-2019. Please note that the revision deadline will expire at 00.00am on this date. If you do not think you will be able to meet this date please let me know immediately.

- 1) A text file of the manuscript (tex, txt, rtf, docx or doc), references, tables (including captions) and figure captions. Do not upload a PDF as your "Main Document".
- 2) A separate electronic file of each figure (EPS or print-quality PDF preferred (either format should be produced directly from original creation package), or original software format)
- 3) Included a 100 word media summary of your paper when requested at submission. Please ensure you have entered correct contact details (email, institution and telephone) in your user account
- 4) Included the raw data to support the claims made in your paper. You can either include your data as electronic supplementary material or upload to a repository and include the relevant doi within your manuscript

5) All supplementary materials accompanying an accepted article will be treated as in their final form. Note that the Royal Society will neither edit nor typeset supplementary material and it will be hosted as provided. Please ensure that the supplementary material includes the paper details where possible (authors, article title, journal name).

Kind regards,

on behalf of Dr Alexander Ophir (Associate Editor) and Professor Kevin Padian (Subject Editor)
openscience@royalsociety.org

Associate Editor Comments to Author (Dr Alexander Ophir):

Dear Dr. Gardner,

I have received the reviews of your revised manuscript from the same reviewers as before. Your paper has clearly benefited from their critiques, yet the reviewers continued to have some concerns that should be quite easily addressed. While they are important, these issues are also relatively minor overall. Reviewer 1 reiterated their concern that the litter sizes are unequal and this must be addressed and justified. It strikes me that this can be dealt with fairly easily by discussing the homoscedasticity for your parametric analyses and whether your models accounted for this variance when they were used. However you choose to address having different sample sizes is, of course, ultimately up to you. Reviewer 2 raised additional questions, to those raised previously. Most of these focus on clarifying some very minor points. Although you have added some information about your R Code, you will want to pay special attention to providing additional clarity as questions persisted. You should also provide clear justification for why you emphasize main effect results when interactions results are also found. Although interaction effects supersede the impact of main effects, and occasionally render main effects less meaningful, there are cases in which main effects can be quite meaningful and informative when interactions exist. You will want to be clear about which situations you have and how these shape your interpretations.

Best Regards
Alex Ophir
Associate Editor - RSOS

Subject Editor comments to the Author (Professor Kevin Padian):

Thanks for your attention to the reviewers' comments. I support the AE's recommendation that we accept your manuscript with the modifications detailed in the attached. Best wishes for your revision.

Reviewer comments to Author:

Reviewer: 2

Comments to the Author(s)

The authors have responded to the majority of comments made on their original submission. However, in our original review, we asked "How many litters were used in the analysis with less than three pups? If data from these is removed, are the differences detected still significant?" The authors have not directly answered this question. Instead, the authors restate that there is no difference in litter size between the 24 conspecific and 17 hybrid litters. However, all 41 litters are not used in every test ex. prepregnancy activity N = 16 and N = 10; retrieval N = 13 and N = 12; post pregnancy activity N = 15 and N = 13; milk let down N = 15 and N = 14 and USVs N = 3 and N = 6. It is important to demonstrate that parity and litter size are balanced for the animals included in each test, not those used overall.

"Likewise, the fact that maternal behaviors are affected by manipulation of either *Peg3* or *Phlda2* in offspring (Curley et al. 2004; Creeth et al. 2018)" implies that Curley et al., 2004 included a study on maternal behaviour in a model where WT dams carry and care for *Peg3* mutant pups. They report on litter size, pup weight gain per se and after separation, and maternal weight gain but did specifically examine maternal behaviour for WT dams that carry and care for *Peg3* mutant pups. Therefore this citation is incorrect.

Reviewer: 1

Comments to the Author(s)

This well-written paper describes a well-conceived study that examined the effects of altered placental signaling on maternal behavior. It compared the behavior of females who had hybrid offspring versus those that had conspecific offspring. The Authors found indirect evidence that the placenta (which previous work showed was reduced when offspring were hybrid) is critical in priming maternal behavior, with mothers of hybrid offspring having reduced maternal responsiveness and being more anxious (at least earlier in pregnancy). The reduction in genotype effect over time suggested that exposure to pups mitigated the effects generated by differences in placental signaling.

I very much appreciate the changes the Authors made in response to my and the other Reviewer's comments. I have detailed comments below, but my major criticisms are still about the description of the statistical analyses. I understand the authors made some attempt to address my previous criticisms of their statistical analyses section. However, I still have strong concerns. In particular, 1) what is said in the text often does not match what is in the included R code, 2) it is not specifically stated in the text whether model assumptions were examined (in particular, I'd like to know whether their data were overdispersed in models where they assumed a Poisson distribution), and 3) LMMs are not consistently analyzed (two different R packages are used, *lme4* and *nlme*, with no explanation of why the switch). These discrepancies make me uncomfortable with the manuscript and I hope they can be addressed.

More specific comments:

Introduction:

Line 77: Would it be worth specifying “maternal” hypothalamus to make it clearer? At least that is my inference; if I am wrong about this, then I STRONGLY advise adding a clarifying descriptor.

Methods:

Lines 160-161: “starting when the mouse first entered the outer edge of the grid.” Can you explain? Did they generally go to the edge first and then later move into the center (I know from below that some of them simply froze...)?

Lines 206: Could you include in the Results how the variables from the open field assay are correlated?

Line 207: Many of your GLMs used a Poisson distribution. Did you ever verify this was appropriate? For example, state that you examined the residuals for normality in the Methods (which you said you did in the response to my previous comments). Also, did you test to see if your data were overdispersed? If so, a Poisson distribution would be inappropriate, although perhaps using a the “quasi-poisson” family would be OK.

Line 207: “the lme4 package” This statement is incorrect given the code you shared (which uses the “glm” function) and the fact that functions in lme4 require a random statement, and you don't have one here.

Lines 209-210: “for pairwise comparisons at each time point” I don't understand what this means. That's not what a repeated-measures ANOVA does if you are modeling time as a continuous variable (if you are not, you should defend this).

Line 210: “using the car package” Something else must have been used to make this model because no function in the car package does this. My guess is that you used car to generate ANOVA tables.

Line 210-211: “with maternal ID and maternal experience included as random effects.” Your R code only has maternal ID in the random statement.

Line 213: “the lme function” Why use lme4 for one set of models and nlme for the other?

Line 215-216: “At time points where maternal activity levels differed” Differed how? How was this determined?

Line 217: “repeated measures ANOVA” Using what function/package?

Line 217: “time” Not in the R code, although it should have been; instead it is included nested within maternal ID in the random statement. It's not clear why this was done here and not in your other repeated-measures ANOVAs.

Figure 1.

I still have some reservations about how the results from the open field assay were interpreted, although I very much appreciate how the authors incorporated my feedback on their previous version. I would re-word the caption to follow the description given in the beginning: “Activity and anxiety-like behaviors”. For example, (a) says “Activity”, whereas (b-d) are presumably “anxiety-like” behaviors? The main comment I have is that I'm a little skeptical of making a big deal about main effects in (a) and (b) when they are not apparent in the figure and there was a significant interaction effect. Because of the significant interaction, it is clearly not the case that one genotype was always higher than the other.

Discussion:

Lines 305-306: “females carrying hybrid litters spent more time in the center of the arena as pregnancy progressed”. What is the earliest day at which the hybrid placenta could be impacting the maternal brain? Because it seems that in all these cases, the hybrid and conspecific moms start out different, but become similar closer to parturition. This result in particular seems to not fit this statement. By parturition, the two genotypes did not appear to differ at all. Rather earlier in pregnancy, mothers with hybrid litters seemed more anxious.

Line 347: "Grb10 and Phlda2" Can you be explicit about the direction of differential expression, rather than just that there are differences? For example, much later in this paragraph it is said that Phlda2 is overexpressed in hybrids. It would be nice to know that sooner.

Line 364: "will focus on isolation of genomic intervals" This is foreshadowing some kind of GWAS or QTL study?

Author's Response to Decision Letter for (RSOS-190732.R1)

See Appendix B.

Decision letter (RSOS-190732.R2)

20-Aug-2019

Dear Ms Gardner,

I am pleased to inform you that your manuscript entitled "Placental genotype affects early postpartum maternal behaviour" is now accepted for publication in Royal Society Open Science.

on behalf of Dr Alexander Ophir (Associate Editor) and Kevin Padian (Subject Editor)
openscience@royalsociety.org

Appendix A

Editor

I have now received reviews from two experts in the field who were both impressed by your study. Each reviewer raised some points that you will need to address before publication, most notably justifying and clarifying your statistical approaches and interpretations from them. I believe addressing these and the other minor points details below should be relatively simple to address. I agree with both of the reviewers that you have conducted an elegant and interesting and well written study that will be an important to the literature once these reviewers comments are addressed.

We thank the editor for their positive assessment and address reviewer comments below and in the revised manuscript. Responses are in blue font, quoted text from the manuscript is italicized, changes are underlined.

Reviewer 1

I found the paper very well written, interesting, and a joy to read, especially the Introduction and Discussion. My major criticisms are 1) a lack of detail regarding the statistical tests (e.g., Why were particular tests used? Which packages/functions in R were used? How were model assumptions evaluated?), and 2) some quibbles with how interaction coefficients are interpreted. Unless the results are strongly affected by addressing these issues (which the figures suggest probably won't be the case), then it will be great to see this study published! I really must emphasize, however, that it is impossible to fully understand how the data was analyzed, or why it was analyzed that way, from what is written.

I give my specific comments below:

Introduction

Lines 72-73: "imprinted genes are predominantly expressed from only one allele". I personally found this wording a little awkward; perhaps "only a single allele of an imprinted gene is predominantly expressed"?

Changed to: *Classified by their unique mode of expression, imprinted genes are autosomal genes with monoallelic expression.*

Lines 79-80: "placental endocrine compartment". I can more-or-less figure out what this means through context, but later in the text (Lines 117-118), an actual definition is given, and it would be helpful if that definition was moved up to here.

Definition moved up as suggested and deleted below.

Line 85: "Females with silenced Peg3". Does this mean neither the paternal nor the maternal allele is expressed? Please clarify.

Peg3 is an imprinted gene that is expressed from the paternally-inherited allele only. If this allele is silenced the gene is not expressed. We have tried to clarify this

in the preceding sentence: *The best-studied example, Peg3 (paternally expressed gene 3) is co-expressed from the paternally inherited allele in the hypothalamus and the placenta, and affects maternal behaviors (Li et al. 1999).*

Lines 82-100: Somewhat related, I was curious whether there have been any experiments expressing the normally non-expressed allele?

To our knowledge, nothing is published on *Peg3* overexpression with regards to behavior in mouse. We found one study that overexpressed the gene in vitro (Johnson et al. 2002 J Biol Chem) but this has no relevance to our study so we didn't add the citation.

Line 102-105: I was curious about these phrases: "natural system" and "misexpressed". What makes this a more "natural system" than other mouse work? Does a hybrid like this ever occur in the wild? These are two strains that were lab created and then crossed, correct? It's a bit of a quibble, but I think there might be something meant here that isn't being fully expressed, and if so, it would be great to have it explicated more clearly. Maybe the real distinction here is between single gene vs. multiple gene manipulations? Related to this, I was curious about the use of "misexpressed". Are these genes expressed the same way in both conspecific crosses? Is it the regulation of expression and/or allele silencing that is disrupted by the hybrid cross?

We have attempted to clarify "natural system" and "misexpressed" as follows: *However, while single gene manipulations can reveal the function of individual imprinted genes, using a natural hybrid system (i.e. species that occasionally hybridize in nature; Liu et al. 2015) in which multiple genes are transgressively misexpressed provides a more holistic view of the impact of imprinted genes on maternal behaviors.*

Methods

Lines 158-161: Was this variation in experience ever utilized in statistical analyses?

The variation in maternal experience was included in the analysis of home cage activity but was not included in the best model. This variation was also included as a random effect in the repeated measures ANOVA for pup retrieval but was not included in the best model.

Lines 177-178: Clarify that anxiety is expressed by not going into the center (or staying near the edge).

We added this information as follows: *Trials were scored for number of lines crossed (a measure of activity and exploration), and latency to first enter the central squares in the grid and total time in center (measures of anxiety-like behaviors, where longer latencies and less time in center are proxies for a more anxious phenotype).*

Line 191: Clarify that all pups were returned to the nest after the test . . .

Clarified as follows: *Litters were only used in this behavioral test if the number of pups in the litter was three or more; additional pups not used in the test were placed under a heat lamp and returned to the home cage at the end of the test.*

Lines 193-198: Can you give some additional information about why this schedule was chosen? Does more suckling happen at night?

Our assay design followed that of Curley et al. (2004). This has been clarified in the text: *Our assay design followed that of Curley and colleagues [25].*

Line 200: Can you be more specific about what “activity levels” means? Is it the distance traveled?

Information added as follows: *Home cage activity levels of females with pups were monitored continuously from parturition for 96 hours (termination was at the onset of the light cycle 5 days after parturition) using an automated monitoring system that recorded the number of times a female crossed an infrared beam per unit time (VitalView Animal Monitoring Software, Version 5.0).*

Lines 218-229: I found that this section was sorely lacking a lot of critical details. None of the packages and functions used were listed (especially for LMMs, this is a problem because you can get different results depending on the package and function used for p-value estimation). Why ANOVAs were used in some cases, but mixed models in others was also not explained. Why at some point a general linear model was used (and what error structure/link function was used) was not described. Effects are sometimes referred to as “fixed” and other times as “explanatory”. Were model assumptions ever checked? The role AIC played in analysis wasn’t fully explained. If an analysis is done post-hoc to further examine a main or interaction effect, this should be clearly stated. Finally, it would be very helpful to present the methods for each assay in the same order as the analyses and likewise, in the same order for the Results. This paragraph for me really sunk an otherwise good paper.

We thank the reviewer for their comments on this section of the paper. We have tried to address these comments and clarify areas that were lacking detail. Information added as follows as related to each comment:

R Package/function: R package information was added for each analysis in the statistical analyses section, together with the error structure for the general linear model.

Using ANOVAs vs. models: ANOVA or repeated measures ANOVA were used when direct comparisons were possible – as in the pup retrieval (hybrid group vs.

conspecific group) and suckling/milk letdown assays – whereas the open field trials were conducted over a range of pregnancy days, requiring a more complex analysis that took the additional information into account. Linear mixed models were chosen for USV production and home cage activity to include the random effects of litter ID and maternal ID, respectively.

Using “explanatory” or “fixed: We now refer to all effects as explanatory variables rather than fixed effects in Statistical Analyses section.

Testing model assumptions: Model assumptions, such as normality of residuals and testing for multicollinearity, were met for the best model for each analysis.

AIC use: AIC use for model selection was further clarified in Statistical Analyses section as follows: *Model selection for open field trials, USV production, and home cage activity was conducted using Akaike Information Criterion (corrected for small sample sizes) where the model with the lowest $\Delta AICc$ value was chosen as the best representative model for the data (AICc tables provided in Supplemental File 1)*

Using ANOVAs for post-hoc analyses: Information clarifying when ANOVAs or other tests were used for post-hoc analyses has been added to the Statistical Analyses section.

Order of statistical methods: Statistical methods were re-ordered in the Statistical Analyses section to match experimental methods and results.

Line 220: It took me some time to realize that “time point” here referred to the first, second, third pup. I find this analysis completely inappropriate as the three times are completely correlated. The time to get the second pup is the time to get the first pup plus the time to get the second pup, for example. Certainly, Figure 2 shows an extremely linear relationship.

We reanalyzed the pup retrieval assay as a repeated measures ANOVA rather than treating each pup as an independent event. This should account for the correlated measures. We made the following change to the Statistical Analyses section: *Pup retrieval results were analyzed using a repeated measures analysis of variance (ANOVA) for pairwise comparisons at each time point using the car package with maternal ID included as a random effect*

We made the following change in the Results section: *Females with hybrid offspring ($n = 12$) took significantly longer to retrieve pups than females with conspecific offspring ($n = 13$) (repeated measures ANOVA: $F_{1,46} = 7.62$ $p = 0.0014$), and significantly longer to retrieve second and third pups (LSM: $p_{\text{Bonferroni-adjusted}} < 0.05$) (Figure 2).*

Line 225-226: The interaction was also included according to the Results.

This has been clarified in the Statistical Analyses section: *Open field trials were analyzed using generalized linear models (GLM) with a Poisson distribution in the lme4 package, with pup genotype, days to parturition, and their interaction as explanatory variables.*

Results

Line 236: Any reason for such a different sample size from the conspecific cross?
Are hybrid litters harder to produce?

Yes, hybrid litters were harder to produce than conspecific litters.

Line 241: The figure makes it appear that hybrid females remain unchanged over time, and it is only the conspecific females that are driving the main effect of activity being reduced towards parturition.

Yes, the graph shows the effect of the interaction more clearly than the individual effects, but the model also included significant individual effects of genotype and time.

Line 244-245: Isn't it more accurate to interpret this positive coefficient as the two lines are becoming more different?

We tried to clarify the different effects with the following: *Females carrying hybrid litters overall spent less time in the center of the arena (estimate ± SE: -3.40 ± 1.06 , $z = -3.19$, $p = 0.0014$), but this time increased as pregnancy progressed, whereas the opposite was true for females carrying conspecific litters (estimate ± SE: 0.180 ± 0.058 , $z = 3.09$, $p = 0.002$, Figure 1b).*

Line 248-249: Again, I think the correct interpretation of the (here negative) interaction coefficient is that the lines become more similar.

We tried to clarify the different effects with the following: *but the interaction between litter genotype and days to parturition showed that females carrying conspecific litters took longer to enter the center of the arena as pregnancy progressed, reducing the difference between the two pregnancy types (estimate ± SE: -0.082 ± 0.032 , $z = -2.59$, $p = 0.0095$, Figure 1c)*

Line 250: "In contrast . . ." I felt that these conflicting results were not explored very much in the Discussion. They were summarized well enough, but I wonder if the authors could speculate more on why there might be this difference.

We agree that the conflicting results are interesting, but further tests of anxiety-like behaviors are necessary to strictly say that mothers carrying hybrids are more or less anxious than mothers carrying conspecific offspring (such as an elevated plus maze, novel object test, and forced swim test). In the absence of these data we thought it best not to speculate on the reasons for the difference.

Line 262: I'm repeating myself, but what is the justification for analyzing each pup retrieval as independent events?

We changed the analysis as suggested. Please see our response to the comment regarding Line 220 (above).

Lines 267-269: Given this sentence, I'm curious how "important" the random effect is. There are a number of different ways "important" could be quantified . . .

This raises an interesting point however we think with additional litters collected, this effect may become more apparent. The intercept value for the random effect was high (StdDev = 29.57), so there seems to be high variability between litters. What we believe is interesting when looking at the results of the USV production is that either there is no difference between pup vocalizations on day one (so differences in mothers alone is driving the differences in pup retrieval) or that the difference is not in the direction expected to impact pup retrieval. One would expect a high rate of USV production to promote faster retrieval. The fact that we find the opposite further supports the inference that postpartum maternal responsiveness is deficient in mothers of hybrids.

Lines 277-288: What are these ANOVAs? Are they some kind of post-hoc comparison of specific time points?

Yes, we added the following to the Statistical Analysis section to clarify this point: *At time points where maternal activity levels differed, the effect of pup genotype was tested with post-hoc ANOVAs.*

Line 298: Is this a main effect of offspring genotype?

This was the result of the repeated measures ANOVA, where offspring genotype was the main effect.

Line 299 and 300: Specify that these are ANOVAs(?)

Clarified as follows: *overall change in mass (repeated measures ANOVA: $F_{1,25} = 0.082$, $p = 0.78$; Figure 4b). However, hybrids lost less weight during the two-hour separation from mothers (ANOVA: $F_{1,72} = 4.27$, $p = 0.042$).*

Reviewer 2

The authors present their study on the impact of hybrid offspring on an exposed dams' behaviour pre and postnatally. They report alterations in anxiety and also pup directed behaviour which, in combination with the bioRxiv submission (Arévalo, L., and P. Campbell. 2019. Placental effects on maternal brain revealed by disrupted placental gene expression in mouse hybrids) lends further support to the hypothesis that signals from the placenta plays a key role in the induction of aspects

of maternal care, and that disruption of imprinting in the placenta can lead to deficits in maternal care via disruption of these signals. The study is elegantly designed and the results are important.

I have some points regarding the litters used in the analysis that require clarification

The authors mention that “Females used in this study were either first (26/41 litters), second (11/41 litters), or third (2/41 litters) time mothers”. They also mention that “Litters were only used in this behavioral test if the number of pups in the litter was three or more”. Both parity and litter size are known to be factors influencing maternal behaviour.

The reviewer raises important points on parity and litter size that we have addressed below:

While parity does affect maternal behavior, we found that maternal experience was not included in the best model for home cage activity or pup retrieval. During breeding for the experiment, WSB females did not differ in average litter sizes based on pup genotype (please see response below).

We only used litters of 3 pups or more in the pup retrieval assay (smaller litters were included in the other behavioral tests) as displacing 3 pups was in the design of the assay in Curley et al. 2004.

How many litters were used in the analysis with less than three pups? If data from these is removed, are the differences detected still significant?

Were there differences in any test in relation to litter size within group? The authors should do a correlation analysis.

Were there differences in any test between first, second and third time dams within group? Anova would be a useful test.

Was the composition of first, second and third time dams balanced statistically across the groups?

Also, the number of litters does not add up to 41?

We have provided additional information to clarify reviewer questions on maternal experience and litter size within groups and across behavioral tests:

Litter size (<3 pups): No litters with less than 3 pups were used for the pup retrieval assay. Litter sizes for open field trials were collected after females gave birth and, in a number of cases, females cannibalized their litter before size was observed. The values for maternal weight change in the suckling/milk letdown assay are very similar despite some variation in litter size within the analysis (litter size range: 2-6 pups). Average litter size was the same between the groups for the home cage activity measure. Overall, litter size was the same between the two groups, which is listed in the mouse husbandry section as follows: *Average litter size did not differ*

between groups (consppecific: 3.78 ± 0.27 pups, hybrid: 3.45 ± 0.23 pups; $t = -0.922$, $df = 40,74$, $p = 0.362$)

Composition of experience balanced across groups: There were more conspecific litters produced than hybrid litters overall. This was also apparent in how many first-time mothers (16 conspecific litters, 12 hybrid litters) and second-time mothers (7 conspecific litters, 4 hybrid litters) there were for each group. Both third-time mothers produced hybrid litters.

Adding up the number of litters: The total number of litters with first time dams was miscalculated and has been changed on the document as follows: *Females used in this study were either first (28/41 litters), second (11/41 litters), or third (2/41 litters) time mothers.*

Minor

Line 25

“how altered placental signaling might affect maternal behavior is unstudied in a natural system”

Could the author rephrase this sentence as this is not the case (Creeth et al 2018)

We think that gene manipulations in lab mice do not constitute a natural system whereas crosses between sympatric species do. We have attempted to clarify what we mean by “natural system” in the main text (please see response to Reviewer 1’s comment re. lines 102-105).

Line 96

“with presumed secondary effects on placental hormone production (Tunster et al. 2016)”.

Not presumed – demonstrated in earlier papers

“Presumed” has been deleted.

Line 97

“Notably, suppression of placental Phlda2 expression in wild-type females increases maternal nursing and grooming of newborn pups, while over expression increases non-pup-directed maternal behaviors such as nest building (Creeth et al. 2018)”.

This sentence needs to be rephrased to be clearer. Creeth et al. reported that loss of function of Phlda2 in the fetally-derived placenta resulted in increased maternal nurturing by wild type dams whereas loss-of-imprinting of Phlda2 (increased expression) resulted in an increased focus of wild type dams on nest building.

Changed at follows: Notably, loss of function of Phlda2 in the fetally-derived placenta results in increased maternal nurturing by wild type dams whereas loss-of-imprinting of Phlda2 (increased expression) results in an increased focus of wild type dams on nest building, a non-pup-directed behavior (Creeth et al. 2018).

Line 101

After the first sentence – include the reviews by Creeth et al 2019 and Potter et al published in Front Neuroendocrinol. 2019 which extensively cover this topic

Citations added.

Line 393

“Likewise, the fact that maternal behaviors are affected by manipulation of either *Peg3* or *Phlda2* in offspring (Curley et al. 2004; Creeth et al. 2018)”

Curley et al. 2004 reports on loss of function of *Peg3* in the dam, not the offspring.

McNamara 2018 reports on loss of function of *Peg3* in the offspring

Curley et al. 2004 studied the effects of silencing *Peg3* in mothers with wildtype offspring and in wildtype mothers with *Peg3* silenced. Please see Fig. 1 and section 2b in their paper. Since then, multiple groups have published on the effects of *Peg3* silencing in mothers or offspring but we think that citing the original study is sufficient.

Appendix B

Associate Editor Comments to Author (Dr Alexander Ophir):

I have received the reviews of your revised manuscript from the same reviewers as before. Your paper has clearly benefited from their critiques, yet the reviewers continued to have some concerns that should be quite easily addressed. While they are important, these issues are also relatively minor overall. Reviewer 1 reiterated their concern that the litter sizes are unequal and this must be addressed and justified. It strikes me that this can be dealt with fairly easily by discussing the homoscedasticity for your parametric analyses and whether your models accounted for this variance when they were used. However you choose to address having different sample sizes is, of course, ultimately up to you. Reviewer 2 raised additional questions, to those raised previously. Most of these focus on clarifying some very minor points. Although you have added some information about your R Code, you will want to pay special attention to providing additional clarity as questions persisted. You should also provide clear justification for why you emphasize main effect results when interactions results are also found. Although interaction effects supersede the impact of main effects, and occasionally render main effects less meaningful, there are cases in which main effects can be quite meaningful and informative when interactions exist. You will want to be clear about which situations you have and how these shape your interpretations.

We thank the editor for their suggestions and address reviewer comments below and in the revised manuscript. Responses are in blue font, quoted text from the manuscript is italicized, changes are underlined.

Reviewer 1

The authors have responded to the majority of comments made on their original submission. However, in our original review, we asked “How many litters were used in the analysis with less than three pups? If data from these is removed, are the differences detected still significant?” The authors have not directly answered this question. Instead, the authors restate that there is no difference in litter size between the 24 conspecific and 17 hybrid litters. However, all 41 litters are not used in every test ex. prepregnancy activity N = 16 and N = 10; retrieval N = 13 and N = 12; post pregnancy activity N = 15 and N = 13; milk let down N = 15 and N = 14 and USVs N = 3 and N = 6. It is important to demonstrate that parity and litter size are balanced for the animals included in each test, not those used overall.

Levene’s tests were conducted on each analysis based on genotype to ensure that variances within the group were not different – this was the case for all parametric analyses. Therefore, we concluded that parity and litter size were balanced in each test. We added the following text (L149-152): *Because a subset of all mothers was used in each behavioral assay, we also used Levene’s test to check for unequal variance in litter size between mothers of conspecific and hybrid litters in a given assay. In all cases the test was non-significant, indicating equal variance across the two groups.*

“Likewise, the fact that maternal behaviors are affected by manipulation of either Peg3 or Phlda2 in offspring (Curley et al. 2004; Creeth et al. 2018)” implies that Curley et al., 2004 included a study on maternal behaviour in a model where WT dams carry and care

for Peg3 mutant pups. They report on litter size, pup weight gain per se and after separation, and maternal weight gain but did specifically examine maternal behaviour for WT dams that carry and care for Peg3 mutant pups. Therefore this citation is incorrect.

This citation has been changed to McNamara et al. 2017 (L357: citation #26) – this study more clearly shows an impact on maternal anxiety than the Curley et al. 2004 paper as the reviewer pointed out.

Reviewer 2

This well-written paper describes a well-conceived study that examined the effects of altered placental signaling on maternal behavior. It compared the behavior of females who had hybrid offspring versus those that had conspecific offspring. The Authors found indirect evidence that the placenta (which previous work showed was reduced when offspring were hybrid) is critical in priming maternal behavior, with mothers of hybrid offspring having reduced maternal responsiveness and being more anxious (at least earlier in pregnancy). The reduction in genotype effect over time suggested that exposure to pups mitigated the effects generated by differences in placental signaling.

I very much appreciate the changes the Authors made in response to my and the other Reviewer's comments. I have detailed comments below, but my major criticisms are still about the description of the statistical analyses. I understand the authors made some attempt to address my previous criticisms of their statistical analyses section. However, I still have strong concerns. In particular, 1) what is said in the text often does not match what is in the included R code, 2) it is not specifically stated in the text whether model assumptions were examined (in particular, I'd like to know whether their data were overdispersed in models where they assumed a Poisson distribution), and 3) LMMs are not consistently analyzed (two different R packages are used, lme4 and nlme, with no explanation of why the switch). These discrepancies make me uncomfortable with the manuscript and I hope they can be addressed.

More specific comments:

Introduction:

Line 77: Would it be worth specifying "maternal" hypothalamus to make it clearer? At least that is my inference; if I am wrong about this, then I STRONGLY advise adding a clarifying descriptor.

This was modified in the text as follows (L76): *in the maternal hypothalamus and the placenta*

Methods:

Lines 160-161: "starting when the mouse first entered the outer edge of the grid." Can you explain? Did they generally go to the edge first and then later move into the center (I know from below that some of them simply froze...)?

Yes, typically mice will quickly run to the edge of the arena. Freezing behavior is not a usual metric of the open field test, but was observed in a number of individuals, so it was then included.

Lines 206: Could you include in the Results how the variables from the open field assay are correlated?

Correlations between open field variables were added to the results section as follows (L241-243): *Most metrics for the open field test were weakly correlated ($|r| < 0.19$). The number of lines crossed and latency to enter the center of the arena were moderately negatively correlated ($r = -0.58$).*

Line 207: Many of your GLMs used a Poisson distribution. Did you ever verify this was appropriate? For example, state that you examined the residuals for normality in the Methods (which you said you did in the response to my previous comments). Also, did you test to see if your data were overdispersed? If so, a Poisson distribution would be inappropriate, although perhaps using a the “quasi-poisson” family would be OK.

Because we considered the variables for this test to be count data, we originally chose the poisson distribution, however we did not check for overdispersion. We thank the reviewer for asking this important question as all of the GLM models showed overdispersion. We therefore checked that the Poisson distribution was appropriate for each variable using the Shapiro-Wilk test. The normal distribution was appropriate for 3 of the variables: number of lines crossed, time in the center of the arena, and latency to enter the center of the arena. This was not the case for the time spent frozen at the onset of the trial, so we chose the negative binomial distribution because it accounts for the overdispersion (the ratio of deviance and df was close to 1) and, unlike the quasi-Poisson, allows for AICc comparison on the models generated.

After re-running the models with different distributions our results for this behavioral assay have changed. The null models are the best fit models for each of the variables that we measured. This has changed the subsection of the Results focused on the open field test (L232-243) and the short portion of the Discussion (L293-303) that dealt with these results but not the main conclusions of the paper.

Line 207: “the lme4 package” This statement is incorrect given the code you shared (which uses the “glm” function) and the fact that functions in lme4 require a random statement, and you don't have one here.

We thank the reviewer for catching this error. In fact we used the nlme package. This has been changed in the text (L211).

Lines 209-210: “for pairwise comparisons at each time point” I don't understand what this means. That's not what a repeated-measures ANOVA does if you are modeling time as a continuous variable (if you are not, you should defend this).

This statement has been removed from the text.

Line 210: “using the car package” Something else must have been used to make this model because no function in the car package does this. My guess is that you used car to generate ANOVA tables.

The car package was used to run the `anova.lme` function to generate the ANOVA tables and the `nlme` package was used to generate models. The latter has been added to the text (L211 & 221).

Line 210-211: “with maternal ID and maternal experience included as random effects.” Your R code only has maternal ID in the random statement.

A second model that included maternal experience was used for comparison. However, the model without experience was chosen based on lower AIC value. This code has been added to the Supplemental Text File 2 (R Code).

Line 213: “the lme function” Why use `lme4` for one set of models and `nlme` for the other?

The models generated for the repeated measures analyses were re-run so that they all used the `nlme` package to generate models for the ANOVA. This did not change the results.

Line 215-216: “At time points where maternal activity levels differed” Differed how? How was this determined?

This was determined visually based on standard errors. We have added this information to the text (L219-220): *At time points where maternal activity levels differed qualitatively (non-overlapping standard errors), the effect of pup genotype was tested with post-hoc ANOVAs.*

Line 217: “repeated measures ANOVA” Using what function/package?

This information is now included in the text as follows (L220-222): *Suckling and milk letdown were analyzed using repeated measures ANOVA with the nlme and car packages, with pup genotype, time, and their interaction as explanatory variables.*

Line 217: “time” Not in the R code, although it should have been; instead it is included nested within maternal ID in the random statement. It's not clear why this was done here and not in your other repeated-measures ANOVAs.

We re-worked the code and this is now accounted for. The null model was the best model in this instance.

Figure 1.

I still have some reservations about how the results from the open field assay were interpreted, although I very much appreciate how the authors incorporated my feedback on their previous version. I would re-word the caption to follow the description given in

the beginning: "Activity and anxiety-like behaviors". For example, (a) says "Activity", whereas (b-d) are presumably "anxiety-like" behaviors? The main comment I have is that I'm a little skeptical of making a big deal about main effects in (a) and (b) when they are not apparent in the figure and there was a significant interaction effect. Because of the significant interaction, it is clearly not the case that one genotype was always higher than the other.

We have added the following clarifications in the figure caption per the reviewer's first point: "*Figure 1. Activity (a) and anxiety-like behaviors (b-d) of females pregnant with hybrid (grey circles) or conspecific (black diamonds) litters in the open field test.*"

Thanks to the reviewer's earlier comment about overdispersion in our GLMs, we have reanalyzed the GLMs with different distributions that have changed the results of this behavioral assay. Because the null models were the best fit models for all of the variables collected, we have changed the Results (L232-243) and Discussion (L289-303) sections to reflect this.

Discussion:

Lines 305-306: "females carrying hybrid litters spent more time in the center of the arena as pregnancy progressed". What is the earliest day at which the hybrid placenta could be impacting the maternal brain? Because it seems that in all these cases, the hybrid and conspecific moms start out different, but become similar closer to parturition. This result in particular seems to not fit this statement. By parturition, the two genotypes did not appear to differ at all. Rather earlier in pregnancy, mothers with hybrid litters seemed more anxious.

The referenced line has been removed. To address the reviewer's question, placental lactogen expression is detected as early as embryonic day 8 through late gestation in mice (Simmons et al. 2008, citation # 11), giving a long period of time in which abnormal placental signals may impact behavior. Additionally, the placenta is fully mature by embryonic day 12, thus still allowing the period of late gestation to be influenced by these signals.

Line 347: "Grb10 and Phlda2" Can you be explicit about the direction of differential expression, rather than just that there are differences? For example, much later in this paragraph it is said that Phlda2 is overexpressed in hybrids. It would be nice to know that sooner.

We have inserted the following sentence on L342-344: *Specifically, Grb10 is upregulated in hybrid placenta relative to M. m. domesticus and Phlda2 is overexpressed relative to both parental species [34].*

Line 364: "will focus on isolation of genomic intervals" This is foreshadowing some kind of GWAS or QTL study?

We have added the following information (L359-363): *Future work on the M. m. domesticus/M. spretus cross will use backcross mapping to identify genomic intervals that influence maternal behavior. This is a promising strategy for assessing the contribution of imprinted genes as most are clustered in the genome and two such clusters contain multiple genes with altered expression in hybrid placenta [34].*